# Perineural Invasion in Pancreatic Ductal Adenocarcinoma (PDAC): A Saboteur of Curative Intended Therapies?

**DOI:** 10.3390/jcm11092367

**Published:** 2022-04-23

**Authors:** Matthäus Felsenstein, Flora Lindhammer, Mathilde Feist, Karl Herbert Hillebrandt, Lea Timmermann, Christian Benzing, Brigitta Globke, Dario Zocholl, Mengwen Hu, Uli Fehrenbach, Bruno Valentin Sinn, Uwe Pelzer, Igor Maximillian Sauer, Johann Pratschke, Thomas Malinka

**Affiliations:** 1Department of Surgery, Campus Charité Mitte/Campus Virchow-Klinikum, Charité-Universitätsmedizin Berlin, Augustenburger Platz 1, 13353 Berlin, Germany; floratheresa.lindhammer@charite.de (F.L.); mathilde.feist@gmx.de (M.F.); karl-herbert.hillebrandt@charite.de (K.H.H.); lea.timmermann@charite.de (L.T.); christian.benzing@charite.de (C.B.); brigitta.globke@charite.de (B.G.); mengwen.hu@charite.de (M.H.); igor.sauer@charite.de (I.M.S.); johann.pratschke@charite.de (J.P.); 2Berlin Institute of Health (BIH), Charité-Universitätsmedizin Berlin, Anna-Louisa-Karsch-Str. 2, 10178 Berlin, Germany; 3Institute of Biometry and Clinical Epidemiology, Campus Charité Mitte, Charité-Universitätsmedizin Berlin, 10117 Berlin, Germany; dario.zocholl@charite.de; 4Department of Radiology, Campus Charité Mitte/Campus Virchow-Klinikum, Charité-Universitätsmedizin Berlin, 10117 Berlin, Germany; uli.fehrenbach@charite.de; 5Institute of Pathology, Campus Charité Mitte, Charité-Universitätsmedizin Berlin, 10117 Berlin, Germany; bruno.sinn@charite.de; 6Medical Department, Division of Hematology, Oncology and Tumor Immunology, Campus Charité Mitte/Campus Virchow-Klinikum, Charité-Universitätsmedizin Berlin, 10117 Berlin, Germany; uwe.pelzer@charite.de

**Keywords:** pancreatic ductal adenocarcinoma, perineural invasion, risk stratification

## Abstract

(1) Background: Perineural invasion (PNI) is a common characteristic of pancreatic ductal adenocarcinoma (PDAC) and is present in most resection margins. We hypothesized that curative pancreatic tumor resection with long-term survival could only be achieved in PNI-negative patients. (2) Material and Methods: A retrospective investigation of PDAC patients who underwent curative-intended surgery during the period 2008 to 2019 was performed at our institution. (3) Results: We identified 571 of 660 (86.5%) resected patients with well-annotated reports and complete datasets. Of those, 531 patients (93%) exhibited tumors with perineural invasion (Pn1), while 40 (7%) were negative for PNI (Pn0). The majority of patients in the Pn1 group presented advanced tumor stage and positive lymph node infiltration. Patients in the Pn0 group showed an improved disease-free and long-term survival compared to the Pn1 group (*p* < 0.001). Subgroup analysis of all R0-resected patients indicated improved long-term survival and disease-free survival of R0 Pn0 patients when compared to R0 Pn1 patients (*p* < 0.001). (4) Conclusion: Our study confirmed that Pn0 improves the long-term survival of PDAC-resected cancer patients. Furthermore, PNI significantly challenges the long-term survival of formally curative (R0) resected patients. We provide new insights into the dynamics of PNI in pancreatic cancer patients which are needed to define subgroups of patients for risk stratification and multimodal treatment strategies.

## 1. Introduction

Across stages, pancreatic duct adenocarcinoma (PDAC) has overall poor prognosis, with 5-year survival rates of approximately 10% [1]. The incidence of PDAC is rising worldwide, making it the second predicted leading cause of cancer-related deaths until 2030 [2]. When patients present with symptoms at the time of diagnosis, the infiltration of surrounding tissues or metastasis to distant organs has likely already occurred. Only a small number of patients are amenable to curative treatment, which currently requires a combination of neoadjuvant-/adjuvant chemotherapy and surgical resection [3]. The key for appropriate treatment allocation prior to the operation is the clinical assessment of local and systemic tumor extension by cross-sectional imaging in accordance with current consensus guidelines [3,4,5]. Following curative resection, the standard multimodal treatment protocol depends on the performance status and tumor biology but is commonly based on six months of adjuvant chemotherapy with either modified FOLFIRINOX, gemcitabine/capecitabine combination or gemcitabine alone [6,7,8,9]. Eligible patients are increasingly receiving preoperative treatment with the idea of early control of occult dissemination [10]. Major benefits possibly lie in the higher completion rates of systemic therapy, in downstaging of borderline-resectable tumors and higher rates of negative margins [11,12,13]. Still, the majority of patients eventually relapse and larger, long-term interventional studies are needed to ultimately discriminate a significant effect of neoadjuvant treatment on overall survival and recurrence rates.

Reasons for treatment failure and high recurrence rates are manifold. Recurrence patterns in patients most often show systemic dissemination (>75%), suggesting micrometastatic disease at time of operation [14,15]. However, up to 24% have primarily local recurrence, which often correlates with margin status or perivascular invasion. We understood that accurate histopathological assessment has a major impact on overall clinical outcome. Tumor size is consistently relevant, as well as nodal status (including number of nodes) and resection margin [16,17,18,19]. Additional information on margin status, such as the microscopic assessment of perivascular, lymphatic and perineural invasion, is regularly included, while their exact prognostic relevance remains controversial [20,21,22,23].

Perineural invasion (PNI) is a characteristic feature of PDAC and is present in 70% to 95% of resected pancreatic carcinomas [20]. Compared to other solid malignancies, PNI has the highest prevalence in PDAC [24]. PNI can also be detected in the early stages of pancreatic carcinoma, suggesting that it may be an early event in cancer progression [25,26]. Nerval invasion is a complex process driven by reciprocal interaction between tumor cells and nerves, resulting in axonal growth and tumor cell invasion [27,28,29]. The process is orchestrated by several signaling molecules secreted by neural cells and tumor cells (reviewed in Ref. [30]). With accumulated knowledge of the clinical impact on patient outcomes, perineural invasion has been included in the seventh edition of the TNM classification, defined as either present or absent [31].

At this point, many clinical studies have examined the prognostic value of the PNI of resected PDAC, both indicating that PNI is associated with early tumor recurrence and decreased overall survival [20,26]. To what extent PNI diminishes the long-term survival of curative (R0) resected patients in combination with current (neo-) adjuvant treatment protocols has only been partially examined. We hypothesized that, in the setting of complete cancer eradication through surgery, long-term survival can only be achieved in PNI-negative patients. As a high-volume pancreatic cancer center, we aim to explore the impact of PNI in a large cohort of PDAC patients who have undergone curative intended surgery.

## 2. Materials and Methods

### 2.1. Study Design

Patients undergoing curative-intended surgical resection for PDAC at the Department of Surgery, Campus Charité Mitte|Campus Virchow, Charité-Universitätsmedizin, Berlin, Germany, between January 2008 and December 2019 were included. Ethical approval was obtained from the institutional ethics committee (EA1/208/12; EA1/188/17). Data was retrospectively analyzed using a prospectively updated database. Prior to surgery, all patients received radiographic staging (pancreas protocol CT-scan) for evaluation of resectability according to current consensus guidelines [4,5,32]. Selected patients with borderline or locally advanced tumors received preoperative chemotherapy. Subsequent surgical procedures included the Kausch–Whipple operation, pylorus-preserving pancreaticoduodenectomy (PPPD), distal pancreatectomy (DP) or total pancreatectomy. Patients with incomplete medical history documentation, R-status, PNI status, tumor stage as well as patients with in-hospital mortality (<30 days survival) were excluded.

### 2.2. Histopathological Records

Pancreatic tumor samples were routinely examined histologically for the presence of PNI. Histopathological analyses included tumor stage according to the 7th and 8th edition TNM classification: Tumor (T), nodes (N) and metastases (M) classification system, grading, microscopic lymphatic involvement, microscopic vascular involvement, perineural invasion and resection margin. For tumor stage comparison of the Pn0 and Pn1 groups, carcinomas were stratified by tumor size (largest diameter) and artery infiltration (truncus coeliacus; arteria mesenterica superior; arteria hepatica communis) into four groups: (1) tumor size: <2 cm; (2) tumor size: 2–4 cm; (3) tumor size: >4 cm; (4) artery infiltration. To compare lymph node involvement between Pn0 and Pn1 groups, tumors were classified by N0 or N+.

### 2.3. Statistical Analysis

For statistical analysis, the statistical software R (The R Foundation, version 4.0.0, Vienna, Austria) was used. Continuous variables are presented as means (range, standard deviation). For categorical variables, simple comparisons between groups were performed based on Fisher’s exact test. Kaplan–Meier curves were estimated for analyses of time-to-event outcomes, such as overall survival and disease-free survival, and the log-rank test was used for subgroup comparison. To control for confounders, Cox proportional hazards regression was applied. Multivariable analysis with PNI as outcome variable was performed via multiple logistic regression. The confidence level was set to 95% for all analyses.

## 3. Results

### 3.1. Patient Cohort

The histopathological reports of resected PDAC patients were screened during the study period 2008 to 2019 (12 years) to identify patients’ perineural invasion (PNI) status. Of the 660 patients with resected PDAC, in 571 cases (86.5%) clinical data and histopathological records were complete. We identified 40 patients (7%) who were Pn0 and 531 patients (93%) who were Pn1. Patients’ characteristics are summarized in Table 1. There were 22 female (55%) and 18 male (45%) patients negative for perineural invasion (Pn0 group), with a median age of 65.5 years (range 35–87). The group of patients positive for perineural invasion (Pn1 group) consisted of 242 female (45.5%) and 289 male (54.5%) individuals with a median age of 67 years (range 31–94). Tumor location in over 70% of both groups was the head of the pancreas (Pn0 group: 70%, Pn1 group: 75.1%), followed by the pancreatic tail (Pn0 group: 17.5%, Pn1 group: 11.8%) and the pancreatic body (Pn0 group: 7.5%, Pn1 group: 8%). Surgical procedures were performed as either pylorus-preserving pancreaticoduodenectomy (Pn0 group: 50%, Pn1 group: 64.6%), the Kausch–Whipple procedure (Pn0 patients: 12.5%, Pn1 patients: 4.5%), distal pancreatectomy (Pn0 group: 22.5%, Pn1 group: 14.3%) or total pancreatectomy (Pn0 group: 15%, Pn1 group: 16.5%).

### 3.2. PNI Correlates with Advanced Tumor Stage and Lymphatic Dissemination

The histopathological status of patients’ tumors in the Pn0 and Pn1 group are summarized in Table 2. For comparison of tumor stages between Pn0 and Pn1 group, tumors were classified by tumor diameter and artery infiltration according to current consensus [16]. In our cohort, 30% of Pn0 patients presented with early-stage pT1 tumors (Pn0, pT1: 30%; Pn0, pT2: 50%; Pn0, pT3: 20%; Pn0, pT4: 0%), while only 10.6% of Pn1 patients presented with pT1 tumors (Pn1, pT1: 10.6%; Pn1, pT2: 56.7%; Pn1, pT3: 28.1%; Pn1, pT4: 4.6%; *p* = 0.007). The majority of Pn1 tumors showed lymph node metastasis (Pn1, pN+: 75.7% versus Pn1, pN0: 24.3%), while lymph node infiltration occurred only in 40% of Pn0 tumors (Pn0, pN+: 40% versus Pn0, pN0: 60%; *p* < 0.001). This also significantly correlated with lymphatic invasion; 44% of the Pn1 tumors were positive for lymphatic invasion (Pn1 L1: 44.1%; Pn1 L0: 55.9%). Only 17.5% Pn0 tumors were positive for lymphatic invasion (Pn0 L1: 17.5%, Pn0 L0: 82.5%; *p* < 0.001). There also appeared to be a detectable difference in the grading of Pn1 tumors compared to Pn0 tumors, with a shift towards less differentiated tumors in the Pn1 group (Pn1 G1: 3%; Pn1 G2: 60.5%; Pn1 G3: 36.5%-Pn0 G1: 10%; Pn0 G2: 67.5%; Pn0 G3: 22.5%; *p* = 0.026). However, vascular invasion and resection margin did not correlate with either Pn0 or Pn1 tumors.

Using a logistic regression model to assess the potential impact of different variables on PNI, only lymph node metastasis was a significant predictor of PNI (OR: 6.4; *p* < 0.01, Table 3). A total of 20% (8/40) of patients in the Pn0 group and 10.5% (58/531) of patients in the Pn1 group received preoperative chemotherapy. Logistic regression indicated no significant impact of preoperative chemotherapy on PNI. Other preoperative parameters, such as body mass index (BMI), history of type I or type II diabetes mellitus and medication with selective or nonselective betablockers, were evaluated for their impact on PNI. There was a significant difference between the BMI of the Pn0 and Pn1 group (*p* = 0.044). CEA levels between Pn0 and Pn1 group also displayed significant differences (*p* < 0.001). However, according to the logistic regression analyses, there was no significant impact of these preoperative factors on PNI.

### 3.3. Pn0 Status Is Associated with Improved Disease-Free and Long-Term Survival

Studying long-term survivors (LTS with survival > 5 years) in our study cohorts, we discriminated a significantly increased number of LTS in the Pn0 group compared to Pn1 patients (*p* = 0.003). The Kaplan–Meier analysis confirmed that patients in the Pn0 group show dramatically improved overall survival compared to patients in the Pn1 group. Median overall survival was 64.9 months in the Pn0 group compared to 18.1 months in the Pn1 group (*p* < 0.001) (Figure 1A). Patients in the Pn0 group also presented with improved disease-free survival compared to patients in the Pn1 group (Figure 1B). Median disease-free survival was 26 months in the Pn0 group compared to 12.9 months in the Pn1 group (*p* < 0.001). Well established variables impacting the patients’ survival, such as resection margin and N-stage, also negatively correlated with overall- and disease-free survival in our cohort (Appendix A). We therefore performed Cox proportional-hazards regression model for analysis of PDAC patients’ characteristics to identify independent variables impacting the DFS and OS (Table 4). R1 (HR: 1.57; *p* < 0.001), N1 (HR: 1.3; *p* = 0.04), PNI (HR: 1.96; *p* = 0.013) and G3 (HR: 1.27; *p* = 0.035) were independent predictors for DFS. R1 (HR: 1.71; *p* < 0.0001), PNI (HR: 2.01; *p* = 0.009), preoperative (HR: 0.6; *p* = 0.034) and postoperative chemotherapy (HR: 0.72; *p* = 0.007) were independent predictors for OS.

### 3.4. PNI Occurs in the Majority of R0 Resected Patients and Diminished DFS and OS

A total of 359/571 patients (62.9%) of the study population had tumor-free resection margins. Subgroup analysis of R0-resected patients indicated that most R0-resected patients presented with nerval invasion (R0 Pn1: 328/359; 91.4% versus R0 Pn0: 31/359; 8.6%). To investigate what impact PNI has on the disease-free survival and long-term survival of curative (R0)-resected PDAC patients, a Kaplan–Meier analysis of R0 patients was performed. Subgroup analysis of all R0-resected patients indicated improved disease-free survival of R0 Pn0 patients compared to R0 Pn1 patients. Median disease-free survival was 27 months in the R0 Pn0 group compared to 15.6 months in the R0 Pn1 group (HR: 2.41; *p* < 0.001, Figure 2B). R0 Pn0 patients also presented with improved overall survival compared to R0 Pn1 patients (Figure 2A). Median overall survival was 20.95 months in the R0 Pn1 group, whereas median survival in the R0 Pn0 group was not reached. We calculated a significantly increased risk for R0 Pn1 patients (HR: 3.21; *p* < 0.001) compared to R0 Pn0 patients.

### 3.5. Postoperative Chemotherapy Improves OS of Pn1 Patients

A total of 353/571 patients (61.8%) of the study population received postoperative chemotherapy: 70% (28/40) of Pn0 patients and 61.2% (325/531) of Pn1 patients. Patients received either mFOLFIRINOX, Gemcitabine, or were enrolled in clinical studies (CONKO-005, CONKO-006, APACT, HEAT). Chemotherapeutic treatments have been summarized in Appendix A.

According to the Kaplan–Meier analysis, patients with Pn1 tumors receiving postoperative chemotherapy showed significantly improved overall survival. Median overall survival was 19.4 months for patients with Pn1 tumors with postoperative chemotherapy compared to 14.1 months for patients with Pn1 tumors without adjuvant chemotherapy (HR: 0.74; *p* = 0.009, Figure 3A). Median disease-free survival was 14.2 months for patients with Pn1 tumors with postoperative chemotherapy compared to 12.9 months for patients with Pn1 tumors without postoperative chemotherapy, not reaching significance (Figure 3B). Within the first 24 months after curative surgery, however, improved DFS was apparent for patients receiving postoperative chemotherapy. There were no significant differences in OS or DFS for patients with Pn0 tumors receiving postoperative chemotherapy (Appendix A).

Subgroup analysis of patients with R0 Pn1 tumors indicated that postoperative chemotherapy improved OS. Median overall survival was 21.8 months for patients with R0 Pn1 tumors with adjuvant chemotherapy compared to 16.6 months for patients with R0 Pn1 tumors without adjuvant chemotherapy, not reaching statistical significance (HR: 0.79; *p* = 0.11, Figure 4A). Median disease-free survival was 16.6 months for patients with R0 Pn1 tumors with postoperative chemotherapy compared to 14.1 months for patients with R0 Pn1 tumors without postoperative chemotherapy (Figure 4B). Again, focusing on first 24 months, improved DFS for R0 Pn1 patients receiving chemotherapy was detected without statistical significance.

## 4. Discussion

The central aim of this study was to investigate the impact of perineural invasion in curatively treated PDAC patients at our institution. PDAC has the highest rates of PNI compared to other solid malignancies and varies across studies from 60 to 95% [20,24]. In our cohort, 93% of all resected PDAC patients showed histologically confirmed PNI.

Our data indicates that PNI generally correlates with advanced tumor stage. Most patients with Pn1 harbored pT2 and pT3 tumors, while 30% of Pn0 tumors were small (pT1). Patients with Pn1 tumors had significantly higher rates of lymph node metastasis and lymphatic invasion than those with Pn0 tumors. Moreover, as underlined by previous reports, the perineural sheath likely serves as a consecutive route for tumor spread to surrounding tissues and lymph nodes [33,34].

In agreement with the findings of the multicenter study of Crippa S et al. characterizing the clinical impact of PNI in a large cohort, PDAC patients with Pn1 display impaired DFS and OS compared to patients with Pn0 PDAC [26]. This appears particularly relevant in the setting of tumor-free resection margins (R0 Pn1) postsurgery, where Hazard increases up to 3.2 compared to R0 Pn0 tumors. R0 resection was achieved in 62.9% of the entire study cohort, while PNI was detected in the majority of R0 tumors (91%). Multivariate Cox regression analyses confirmed that PNI was an independent predictor of worse DFS and OS, to similar extent as with positive margin (R1) and adjuvant chemotherapy. In consideration of recent studies and meta-analyses about the clinical impact of PNI, this subcohort requests particular attention for improving adjuvant treatment algorithms of affected individuals [20,26,35]. We therefore sought to investigate how PNI impacted the patients’ survival in correlation with current (neo-)adjuvant treatment protocols.

Current standard treatment for resectable PDACs usually includes surgery followed by adjuvant chemotherapy [3]. In our institutional study cohort, low adherence of patients to adjuvant treatment protocols (61.2%) is likely due to decentralized oncologic patient registries (Germany), which challenges the collection of complete information on administered systemic therapies after surgery. Still, we discriminated that adjuvant chemotherapy improved the clinical outcome of PDAC patients with Pn1, significantly impacting their OS and DFS. Subgroup analyses confirmed effects on patients with negative resection margins (R0 Pn1), whereas patients without perineural invasion (Pn0) did not significantly benefit from adjuvant treatments. Most critically, the prevalence of long-term survivors was significantly increased in patients with Pn0 compared to Pn1 (10% vs. 2,8%). Altogether, our study not only adds to the existing evidence of PNI as an independent negative prognosticator on patients’ survival after surgery, but demonstrates that long-term survival is mainly achieved in patients without histological signs of perineural invasion [20,26]. It still remains to be evaluated whether second-line systemic treatments (nonresponders) have an impact on clinical outcome with respect to perineural invasion status [36]. 

Considering current trends in the oncologic treatment of PDAC patients, we also investigated the effects of preoperative treatment on survival regarding PNI status. Currently, clinical decision making is based on preoperative imaging and multidisciplinary assessment, assigning patients to upfront surgery or systemic therapies [37]. During the entire study period, 11.6% of the patients received neoadjuvant treatment. There were no significant differences in the prevalence of PNI between preoperative treatment compared to surgery upfront. Similar results were also reported by recent studies and have been explained by an observational bias, which included patient cohorts of the “pre-neoadjuvant-“ and “pre-FOLFIRINOX” era, a bias we have to consider in our study as well [26,38]. However, some studies were able to discriminate lower rates of PNI in patients with neoadjuvant chemotherapy, underscoring the potential of neoadjuvant protocols in current treatment strategies [39,40].

Nerval invasion is a complex process and the responsible molecular mechanisms have not been sufficiently elucidated. A specific treatment strategy to directly alter the perineural invasion is currently not available. We believe that further research in this field is needed so that PNI can be addressed by targeted treatment modalities which could additionally impact the survival of patients.

We are aware that our study has limitations. As an experienced high-volume pancreatic cancer center, our study cohort includes a large enough dataset to present patients’ demographics in line with reports from other centers. However, our retrospective, single-center study misses out on other reference centers that could offer locoregional variations in patient cohorts and possible differences of pathologic reporting for perineural invasion. In addition, due to trends with changes in the treatment protocols for PDAC over the past decade, our data likely underlies an observation bias, while total numbers were needed to power our study. We therefore focused on valid and robust statements at the cost of detailed subgroup analyses in an effort to minimize type-two errors.

In conclusion, our study confirms that PNI is a common feature of PDAC and correlates with advanced tumor stage and lymphogenic tumor spread of PDAC. PNI occurs in most curative (R0)-resected pancreatic tumors and diminished long-term survival of formally curative (R0)-resected patients. In turn, Pn0 is associated with improved long-term survival of patients after resection. Based on our study, PNI presents a robust and independent predictor of aggressive tumor biology of curative (R0)-resected PDAC patients. Further studies exploring the dynamics of PNI in pancreatic cancer patients are needed to define subgroups of patients for risk stratification and alternative multimodal treatment strategies.

## Figures and Tables

**Figure 1 jcm-11-02367-f001:**
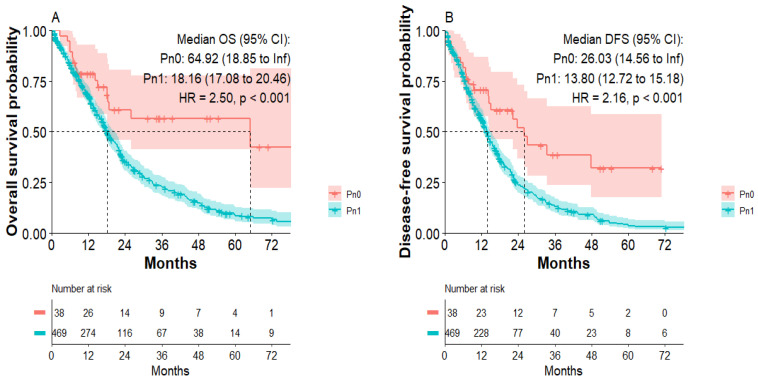
Overall survival and disease-free survival of patients in the Pn0 and Pn1 group. (**A**) Overall survival of Pn0 and Pn1 group. (**B**) Disease-free survival of Pn0 and Pn1 group.

**Figure 2 jcm-11-02367-f002:**
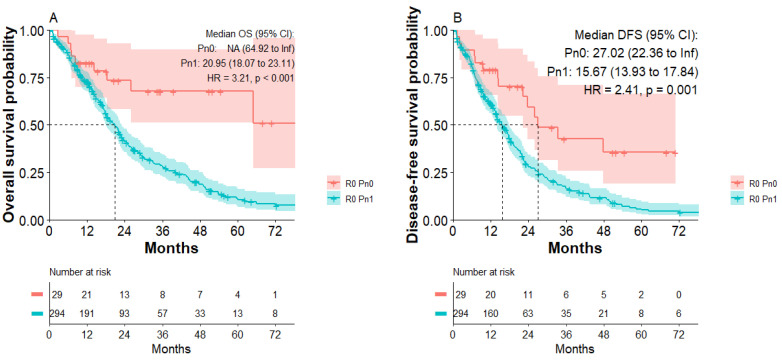
Overall survival and disease-free survival of patients with R0 Pn0 and R0 Pn1 tumors. (**A**) Overall survival of R0 Pn0 and R0 Pn1 group. (**B**) Disease-free survival of R0 Pn0 and R0 Pn1 group.

**Figure 3 jcm-11-02367-f003:**
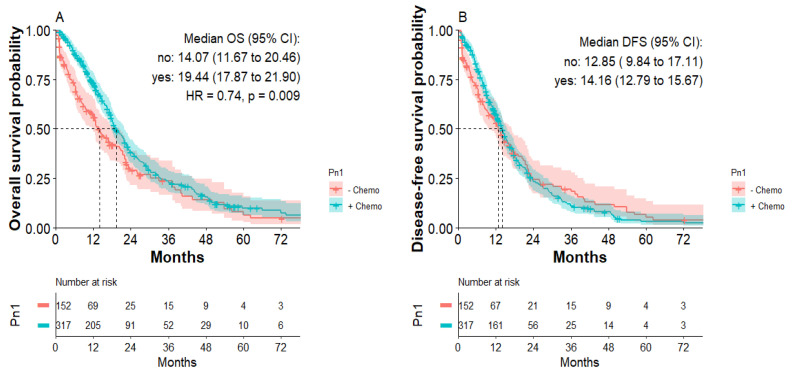
Overall survival and disease-free survival of patients with Pn1 tumors with or without postoperative chemotherapy. (**A**) Overall survival of Pn1 patients with or without postoperative chemotherapy. (**B**) Disease-free survival of Pn1 patients with or without postoperative chemotherapy.

**Figure 4 jcm-11-02367-f004:**
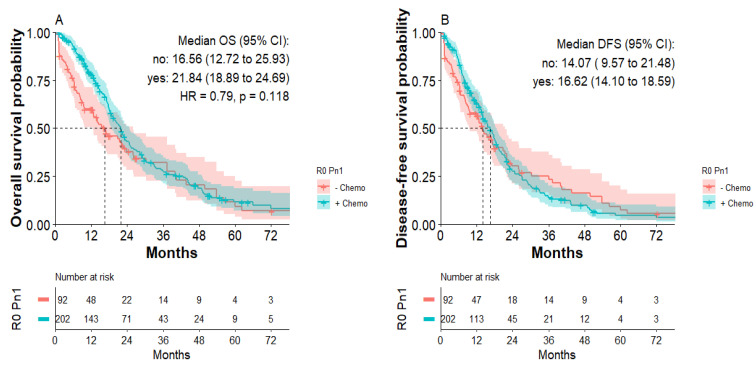
Overall survival and disease-free survival of patients with R0 Pn1 tumors with or without postoperative chemotherapy. (**A**) Overall survival of patients with R0 Pn1 tumors with or without postoperative chemotherapy. (**B**) Disease-free survival of patients with R0 Pn1 tumors with or without postoperative chemotherapy.

**Table 1 jcm-11-02367-t001:** Patient characteristics of entire study cohort, Pn0- and Pn1 group.

	Total	%	Pn0	%	Pn1	%	Statistics *
**Cases**	571		40		531		
**Age** (years)	65.6 (+/−11.1)		64.7 (+/−10.5)		65.6 (+/−10.8)		*p* = 0.56
**Sex**	female 264	46.2	female 22	55	female 242	45.6	*p* = 0.26
	male 307	53.8	male 18	45	male 289	54.4	
**Body Mass Index** (BMI) (in kg/m^2^)	25.1 (+/−4.2)		23.6 (+/−4.0)		25.2 (+/−4.3)		***p* = 0.04**
**Diabetes mellitus** (DM)							
DM I	19	3.3	0	0	19	3.6	*p* = 0.39
DM II	116	20.3	7	17.5	109	20.5	*p* = 0.84
**Beta blocker**							*p* = 0.4
** **ß1 selective	168	29.4	9	23.1	159	31.7	
Non-selective	12	2.1	0	0	12	2.4	
**Carbohydrate-Antigen 19-9** (in U/mL)	846.3 (+/−2946)		453.6 (+/−1303)		888.5 (+/−3142)		*p* = 0.19
**Carcinoembryonic Antigen** (in µg/L)	18.8 (+/−67.9)		4.5 (+/−3.2)		20.0 (+/−72.9)		***p* < 0.001**
**Tumor entity**							*p* = 0.72
Head	427	74.8	28	70	399	75.1	
Tail	70	12.6	7	17.5	63	11.9	
Body	46	6.1	3	7.5	43	8.1	
Uncinate	28	4.9	2	5	26	4.9	
**Surgical procedure**							*p* = 0.06
PPPD **	363	63.6	20	50	343	75.1	
Whipple	29	5	5	12.5	24	4.5	
Total	94	16.5	6	15	88	16.5	
Distal	85	14.9	9	22.5	76	14.3	
**Chemotherapy**							
Pre-operative	66	11.6	8	20	58	10.5	*p* = 0.112
Post-operative	353	61.8	28	70	325	61.2	*p* = 0.311
**Clinical Outcome**							
Follow-up (months)	19.1		24.7		17.1		** *p* ** **= 0.007**
30-day mortality	65	12.3	2	5	63	11.9	*p* = 0.1
Death	383	77.8	15	37.5	368	69.3	*p* = 0.11
Alive	103	20.8	21	52.5	82	15.4	** *p* ** **< 0.001**
Lost to follow up	82	17	4	10	81	15.3	*p* = 0.17
LTS *** (>5years)	19	3.3	4	10	15	2.8	** *p* ** **= 0.003**
Recurrence							
yes	155	27.1	9	22.5	146	27.5	*p* = 0.05
no	416	72.9	31	77.5	385	72.5	

* Fisher’s exact test for categorical variables and students *t*-test for continuous variables. ** Pylorus preserving pancreaticoduodenectomy. *** Longterm survivors (>5years).

**Table 2 jcm-11-02367-t002:** Histopathological assessment of tumors in Pn0 and Pn1 groups.

	Total	%	Pn0	%	Pn1	%	Statistics *
**Cases**	571		40		531		
**Tumor stage**							***p* = 0.007**
pT1 (<2 cm)	68	11.9	12	30	56	10.6	
pT2 (2–4 cm)	323	56.6	20	50	299	56.7	
pT3 (>4 cm)	156	27.3	8	20	148	28.1	
pT4 (vessel infiltration)	24	4.2	0	0	24	4.6	
**Lymphnode metastasis**							***p* < 0.001**
N-	153	26.8	24	60	129	24.3	
N+	418	73.2	16	40	401	75.7	
**Distant metastasis**							*p* = 0.296
M0	510	89.3	38	95	472	88.9	
M1	61	10.7	2	5	59	1.1	
**Histologic grade**							***p* = 0.026**
G1	20	3.5	4	10	16	3	
G2	348	60.9	27	67.5	321	60.5	
G3	203	35.6	9	22.5	194	36.5	
**Resection margin**							*p* = 0.06
R0	359	62.9	31	77.5	328	61.7	
R1	212	37.1	9	22.5	203	38.2	
**Vascular invasion**							*p* = 0.09
V0	467	81.8	37	92.5	430	81	
V1	104	18.2	3	7.5	101	19	
**Lymphatic invasion**							***p* = 0.001**
V0	330	57.8	33	82.5	297	55.9	
V1	241	42.2	7	17.5	234	44.1	

* Fisher’s exact test for categorical variables and students *t*-test for continuous variables.

**Table 3 jcm-11-02367-t003:** Logistic regression analysis of variables and their impact on PNI.

Variable	Odds Ratio	Standard Error	Estimate	*p*-Value
Age	1.155	0.323	0.444	0.657
Sex	0.615	0.684	−0.710	0.478
Tumor size (<4 cm vs. >4 cm)	0.747	0.736	−0.395	0.693
G1-2 vs. G3	1.921	0.755	0.864	0.388
R0 vs. R1	3.166	0.892	1.292	0.196
pN+ vs. pN-	6.080	0.679	2.659	**0.008**
neoadjuvant Chemo	0.830	0.705	−0.264	0.792
BMI	1.517	0.315	1.325	0.185
DM II	0.480	0.813	−0.905	0.366
CEA	5.105	2.140	0.762	0.446
CA 19-9	0.924	0.285	−0.278	0.781

**Table 4 jcm-11-02367-t004:** Multivariate cox regression analysis of variables and their impact on OS and DFS.

Variable	Hazard Ratio	Standard Error	Estimate	*p*-Value
Overall survival (OS)				
Age	1.021	0.005	3.902	**<0.001**
Sex	0.827	0.109	−1.739	0.082
Pn0 vs. Pn1	2.013	0.270	2.596	**0.009**
Tumor size (<4 cm vs. >4 cm)	1.203	0.116	1.591	0.112
pN+ vs. pN-	1.190	0.130	1.340	0.180
R0 vs. R1	1.708	0.115	4.637	**<0.001**
G1-2 vs. G3	1.211	0.112	1.706	0.088
adjuvant Chemo	0.721	0.121	−2.696	**0.007**
neoadjuvant Chemo	0.601	0.240	−2.121	**0.034**
Disease-free survival (DFS)				
Age	1.014	0.005	2.712	**0.007**
Sex	0.872	0.109	−1.261	0.207
Pn0 vs. Pn1	1.963	0.271	2.490	**0.013**
Tumor size (<4 cm vs. >4 cm)	1.204	0.117	1.592	0.111
pN+ vs. pN-	1.308	0.130	2.058	**0.040**
R0 vs. R1	1.568	0.116	3.885	**<0.001**
G1-2 vs. G3	1.269	0.113	2.112	**0.035**
adjuvant Chemo	0.936	0.123	−0.538	0.590
neoadjuvant Chemo	0.658	0.241	−1.737	0.082

## Data Availability

Not applicable.

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
