# Peer review of "Perineural Invasion in Pancreatic Ductal Adenocarcinoma (PDAC): A Saboteur of Curative Intended Therapies?"

_jcm, 2022, doi:10.3390/jcm11092367_

Round 1

Reviewer 1 Report

I appreciate the opportunity to review the authors' work. Please consider several comments.

Of the cases studied, Pn1 is disproportionate for percentage of cases. Is this typical in other studies, the percentage? Can you reference how common this actually is to be this imbalanced?

You note that the majority of Pn1 are also higher stage. We know that higher stage cancers have poorer outcomes. Can you really conclude that the poorer outcomes are due to the Pn1 status or is the true finding that the high stage means poor outcome? 

Reviewer 2 Report

The authours attempted to retrospectively assess the prognostic role of perineural invasion in a signle center study of patients who underwent surical resection. The authors showed that perineural invasion is an independent predictor for worse DFS and OS.

1 The method paragraph lack of information concernig pre-operative staging. Radiographic staging is key for evaluating patients before surgical procedure. Only 64 patients received neoadjuvant treatment. Therefore is relevant to understand the creteria of resectability.

Moreover please add and comment the following manuscript in the discussion:

Vernuccio F, Messina C, Merz V, Cannella R, Midiri M. Resectable and Borderline Resectable Pancreatic Ductal Adenocarcinoma: Role of the Radiologist and Oncologist in the Era of Precision Medicine. Diagnostics (Basel). 2021 Nov 22;11(11):2166

-38% of patients Pn1 did not received adjuvant treatment resulting in poorer outcome. The author should explain about the choice to avoid post-operative treatment.

-Adjuvant strategy was not standardized. Hence the type of regimen may have an influence on outcome

-subsequent treatment may have an impact on final utcomae. Please add and comment the following reference:

Merz V, Cavaliere A, Messina C, Salati M, Zecchetto C, Casalino S, Milella M, Caffo O, Melisi D. Multicenter Retrospective Analysis of Second-Line Therapy after Gemcitabine Plus Nab-Paclitaxel in Advanced Pancreatic Cancer Patients. Cancers (Basel). 2020 Apr 30;12(5):1131.

Round 2

Reviewer 1 Report

I appreciate the authors' consideration and responses to my comments.